# SMRT Sequencing Enables High-Throughput Identification of Novel AAVs from Capsid Shuffling and Directed Evolution

**DOI:** 10.3390/genes14081660

**Published:** 2023-08-21

**Authors:** Widler Casy, Irvin T. Garza, Xin Chen, Thomas Dong, Yuhui Hu, Mohammed Kanchwala, Cynthia B. Trygg, Charles Shyng, Chao Xing, Bruce A. Bunnell, Stephen E. Braun, Steven J. Gray

**Affiliations:** 1Department of Pediatrics, University of Texas Southwestern Medical Center, Dallas, TX 75390, USAirvin.garza@utsouthwestern.edu (I.T.G.); xin.chen@utsouthwestern.edu (X.C.); yuhui.hu@utsouthwestern.edu (Y.H.); 2Graduate School of Basic Biomedical Sciences, University of Texas Southwestern Medical Center, Dallas, TX 75390, USA; 3Eugene McDermott Center for Human Growth & Development, University of Texas Southwestern Medical Center, Dallas, TX 75390, USA; mohammedmsk@gmail.com (M.K.);; 4Tulane National Primate Research Center, Tulane University School of Medicine, Covington, LA 70433, USAbruce.bunnell@unthsc.edu (B.A.B.); sbraun@tulane.edu (S.E.B.); 5Gene Therapy Center, University of North Carolina, Chapel Hill, NC 27599, USA; cshyng@gmail.com; 6Department of Bioinformatics, University of Texas Southwestern Medical Center, Dallas, TX 75390, USA; 7Department of Molecular Biology, University of Texas Southwestern Medical Center, Dallas, TX 75390, USA; 8Department of Neurology and Neurotherapeutics, University of Texas Southwestern Medical Center, Dallas, TX 75390, USA

**Keywords:** AAV, capsid engineering, capsid shuffling, directed evolution, SMRT sequencing, vector development, gene therapy

## Abstract

The use of AAV capsid libraries coupled with various selection strategies has proven to be a remarkable approach for generating novel AAVs with enhanced and desired features. The inability to reliably sequence the complete capsid gene in a high-throughput manner has been the bottleneck of capsid engineering. As a result, many library strategies are confined to localized and modest alterations in the capsid, such as peptide insertions or single variable region (VR) alterations. The caveat of short reads by means of next-generation sequencing (NGS) hinders the diversity of capsid library construction, shifting the field away from whole-capsid modifications. We generated AAV capsid shuffled libraries of naturally occurring AAVs and applied directed evolution in both mice and non-human primates (NHPs), with the goal of yielding AAVs that are compatible across both species for translational applications. We recovered DNA from the tissues of injected animal and used single molecule real-time (SMRT) sequencing to identify variants enriched in the central nervous system (CNS). We provide insights and considerations for variant identification by comparing bulk tissue sequencing to that of isolated nuclei. Our work highlights the potential advantages of whole-capsid engineering, as well as indispensable methodological improvements for the analysis of recovered capsids, including the nuclei-enrichment step and SMRT sequencing.

## 1. Introduction

Adeno-associated viruses (AAVs) are a favored vector for gene therapy, due to their relatively low pathogenicity and high efficiency for gene delivery [1]. Over the last decade, AAV vectors have been increasingly used in clinical trials for nervous system disorders, particularly by injecting the vectors intravenously or into the cerebrospinal fluid (CSF) [2]. AAV-mediated gene transfer has taken advantage of the vector’s inherent safety profile by gutting the AAV of its endogenous genome. Briefly, recombinant AAV genomes replace the AAV gene coding sequences with any DNA less than approximately 4.5 kb, flanked by two inverted terminal repeats (ITRs). The ITRs allow for successful packaging inside of the AAV capsid to generate a recombinant AAV (rAAV) [3]. While regulatory elements within the vector genome can restrict expression to certain cell types, the capsid itself dictates the overall cellular and tissue tropism, as well as the overall efficiency of gene transfer.

In contrast, the architecture of the wildtype AAV genome consists of two open reading frames for Rep and Cap, flanked at both ends by ITRs in a single stranded DNA confirmation of both plus and minus strand conformations [4,5,6]. Rep encodes for proteins that are responsible for replication [7,8], while the Cap gene provides the necessary sequence information to translate the three proteins that form the capsid. The capsid proteins are all translated within the same open reading frame by means of alternative splicing, yielding, from largest to smallest, VP1, VP2, and VP3 [9,10]. A single virion is comprised of 60 VP subunits and they assemble in a stoichiometry of 1:1:10 (VP1:VP2:VP3) to form a T = 1 icosahedral capsid structure of roughly 20–25 nm in diameter [11,12,13,14]. Viral assembly is reported to occur in the nucleolus and is facilitated by assembly activated proteins (AAP and MAAP), which are also encoded within an alternative open reading frame of the cap gene [15,16]. Extensive structural and biochemical analyses of different AAV capsid variants have revealed a highly conserved “parvo coat” at the core of each subunit, comprised of two sets of four alternating β-strands (βB-βI) and an α-helix (αA), also referred to as the jelly roll fold. The subunits also contain unstructured loop regions that have implications in various functional aspects of the capsid, such as cell surface receptor recognition, antigenicity, capsid assembly, genome packaging, and transduction [17,18,19,20,21,22,23,24,25,26,27,28,29,30,31,32]. A recent study that completed the capsid structures of known AAV serotypes found that, while their surface loop conformation differed, the core structural features remained highly conserved [33].

Among the naturally occurring AAV serotypes known to date, AAV9 has been used extensively in pre-clinical studies and in the clinic for the treatment of neurological disorders [34,35,36]. AAV9’s unique properties allow it to cross the blood–brain barrier (BBB) and target CNS tissues for gene delivery [37,38]. Another unique advantage of this vector is its ability to maintain high efficacy across different animal species, including rodents, non-human primates (NHPs), and humans. Altogether, these remarkable features have established AAV9 as a benchmark for newly engineered AAV capsids for CNS gene delivery. AAV9 has been shown to have advantages and differences among varying routes of administration. Preclinical studies of AAV9 CNS gene therapy include systemic (intravenous, IV), intrathecal (IT), intraparenchymal, intra-cisterna magna (ICM), and intracerebroventricular (ICV) routes of administration (reviewed in [39]). IT delivery has been shown to be a safe and effective means of delivering AAV to the CNS through the CSF [34]. Exploitation of these delivery routes is an attractive path for enhancing the efficacy of AAV gene therapies.

AAV capsid engineering for CNS targeted applications using capsid shuffling and directed evolution has proven capable of generating novel capsids, albeit with certain significant challenges. One caveat has been the relatively low throughput analysis of libraries and assembly of recovered sequences using Sanger sequencing technology or first-generation sequencing. An advancement in technology, the emergence of next generation sequencing (NGS), has enabled a higher throughput when shorter sequencing reads are an option, such as peptide insertion libraries. However, sequencing options remain limited for libraries containing whole-capsid manipulations (i.e., random mutagenesis or capsid gene shuffling). This shortcoming can be overcome by using third generation sequencing technology, single molecule real-time (SMRT) sequencing, which is a high-throughput sequencing technology that leverages zero mode waveguides (ZMWs) coupled with 5′ phosphate labeled nucleotides to deliver long reads of individual DNA duplexes [40,41,42]. The SMRT sequencing methodology can facilitate high-throughput analysis of entire AAV libraries, as well as newly recovered pools of capsid variants.

Another major challenge has been highlighted in recent reports of engineered vectors that do not retain cross-compatibility across different animal species, as was seen previously in the case of AAV-PHP.B [43]. AAV-PHP.B was shown to target the brain of C57Bl/6J mice at roughly 40-fold higher efficiency than AAV9. However, subsequent studies have since demonstrated that this high performance is mouse strain and animal species dependent [43,44,45,46]. In a different study, the enhanced version of AAV-PHP.B (“PHP.eB”) was shown to have even greater CNS transduction within C57Bl/6J [47]. Since then, a subsequent study similarly demonstrated that the enhancement not only depends on the route of administration but also the mouse strain [48]. A potential solution to the cross-compatibility challenge is to perform the selection in different animal species, such as mice and NHPs, to help pinpoint engineered AAVs that are compatible across different species. Therefore, refinement in screening selection strategies coupled with high-throughput bioinformatic analysis of long read sequencing can establish a pipeline for generating novel AAVs that span whole-capsid permutations.

Our main goal was to develop AAV vectors that are compatible across mice and NHPs for the translation of CNS gene therapy applications. These libraries were engineered using capsid shuffling and directed evolution in parallel administration using mice and NHPs. AAV capsid libraries were injected via IT-lumbar puncture. The first round of selection consisted of viral genomes recovered from various bulk tissue samples and subjected to SMRT sequencing analysis, to determine the identity of the recovered clones. Four newly identified capsids (P126, P295, P72, and P2909) were chosen from this first round of selection, to assess their transgene delivery efficacy in mice. Immunohistochemistry studies indicated that P126 and P295 showed specificity towards CNS tissues, while clones P72 and P2909 displayed a much broader tissue distribution. Total viral genomes that were recovered from mice and NHP brain regions were used to generate two new libraries, in order to perform a second round of selection. In the second round of selection, the library derived from NHPs was injected (IT) into mice and the library derived from mice was injected (IT) into NHPs. In contrast to the first round of selection, SMRT sequencing was used to identify viral genomes that were purified from isolated nuclei of brain tissue samples rather than bulk tissue samples, in order to further select for clones that efficiently traffic to the nuclei of these tissues. SMRT sequencing data highlighted major differences in the composition of the capsids selected via bulk tissues selection vs. nuclei isolation selection. Overall, we demonstrate an approach to AAV capsid engineering via capsid shuffling and directed evolution that is capable of high-throughput whole-capsid sequence identification using SMRT sequencing technology.

## 2. Materials and Methods

### 2.1. Capsid Library Preparation and Selection in Mice and NHPs

The starting library was generated in a two-phase process as was conducted previously [48,49,50]. The first phase was the plasmid DNA library preparation, and the second phase was the AAV library production. The starting plasmid library was created by shuffling the parent AAV capsid genes of AAV1, AAV2, AAV6, AAV8, AAV9, AAV Olig001, AAV2i8, AAV 9.47, AAV clone 32, AAV clone 83, AAV clone 114, AAV-Rh10, AAV clone 1, and AAV2.5 [49,51,52,53,54,55]. The shuffling process was performed as previously [49]. In brief, DNase I enzyme from New England Biolabs (NEB) was used to randomly digest a DNA mixture containing equimolar amounts of the parent capsid genes and these were reassembled. Subsequently, several rounds of error-prone PCRs were performed using a low fidelity *Taq* polymerase to introduce additional mutations and increase the diversity of the capsid gene pool. Particular care was taken to avoid over-representation of the AAV8 C-terminus, which is similar between AAV Olig001 and AAV clones 1, 32, 83, and 114, by amplifying and shuffling only the N-terminus of these into the mixture with only 1 representative equimolar copy of the C-terminus. The PCR product was then cloned into the pSSV9 plasmid backbone, substituting the existing AAV2 cap gene to create the final plasmid library.

The final shuffled AAV library was created in 2 steps, to maximize the probability of each AAV genome packaging within its own capsid, as well as to minimize the formation of chimeric capsids. In the first step, suspension HEK293 cells were transfected with pALD-X80, as well as the shuffled library plasmid stock and pXR2 (containing Rep2 and Cap2). The library plasmid stock and pXR2 were mixed at a 1:10 ratio to produce an excess of AAV2 capsids during AAV production, packaging each shuffled AAV genome into a mostly AAV2 capsid. The AAV2 shuttle library was then co-infected with wild-type adenovirus into HEK293 cells at a multiplicity of infection of 0.5 vg/cell and 5 infectious units/cell, respectively, to generate the final shuffled AAV capsid library. The WT adenovirus was purchased from the Vector Development Lab at the Baylor College of Medicine. Note that, although the final library production was the same for all three libraries, the 2nd round mouse-derived library and the 2nd round NHP-derived library originated from the original capsid library, which was injected in mice and in NHPs. The viral genomes recovered from the total CNS tissue lysate of these animals were used to make these libraries. In order to assess the cross-compatibility of the recovered capsids, the mouse-derived library was injected in the NHPs and the NHP-derived library was injected in the mice. All injections were performed via the IT route of administration. NHP library injections were performed with approximately 4 × 10^13^ vg per animal (1.5 mL), and mouse library injections were performed with approximately 1.4 × 10^11^ vg per animal (5 µL).

The mouse-derived library and the NHP-derived library were generated similarly to the original library, except that the DNA duplexes that were cloned in the SSV9 backbone were derived from the viral genome mixture that was recovered from the mouse and the NHPs, respectively. This recovery process is explained in later sections.

### 2.2. Library Selection and Recovery

All animal studies carried out at the University of Texas Southwestern Medical Center and the Tulane University School of Medicine were in compliance with their respective Institutional Animal Care and Use Committee (IACUC). The original library was injected in one male mouse and one female mouse, and in one male and one female rhesus macaque, via a lumbar IT route of administration. Two weeks post injection, the animals were euthanized, and the viral genomes were recovered from bulk CNS (whole brain or spinal cord for mice, discreet spinal cord and brain regions for NHPs) and peripheral organ tissue samples. In contrast to the first round of selection, the second round of selection was carried out using single nuclei of the tissue samples instead of bulk tissue, to increase the probability of recovering AAV clones effectively trafficked to the nucleus. The nuclei isolation was performed using a Nuclei Isolation Kit: Nuclei EZ Prep (Sigma Aldrich, St. Louis, MO, USA), according to the manufacturer’s guidelines. Using either bulk tissue or purified nuclei, total DNA (including AAV library genomes) was recovered using a Qiagen DNeasy Blood and Tissue extraction kit.

### 2.3. DNA Preparation for SMRT Sequencing Analysis

The library capsid genes were recovered from mouse or NHP DNA samples following selection via PCR using the forward (CAATAAATGATTTAAATCAGGTATGGCTGCCG) and reverse (GCCGGCTCTAGACGACACCAAAGTTCAACTG) primers, which matched the external sequences present in all AAV genomes in the library. Each reaction was carried out using a Velocity polymerase reaction kit from Bioline, in accordance with the manufacturer’s instructions. The final PCR product for each sample was purified through agarose gel electrophoresis, followed by a Zymoclean Gel DNA Recovery Kit (Zymo Research, Murphy Ave, CA, USA), then submitted to the UTSW McDermott Center for Human Growth and Development Sanger Sequencing Core for processing. The DNA duplex PCR products underwent DNA damage repair, followed by addition of ligation of the adaptors at each terminus using a Template Prep Kit (Pacific biosciences, Menlo Park, CA, USA). The adaptor-Cap DNA complexes were purified using a MagBead Kit from Pacific Biosciences, according to the manufacturer’s protocol. Size selection and additional purification were carried out using an AMPure^®^ PB Kit, which was also from Pacific Biosciences. Subsequently, primers and polymerase from the DNA/Polymerase Binding Kit (Pacific biosciences) were added to the mixture according to the manufacturer’s guidance, and the sample was transferred to the Pacbio Sequel instrument for the SMRT sequencing reaction. The reactions were carried out in SMRT Cell 8Pac from Pacific Biosciences in the presence of SMRT Cell Oil, which was also from Pacific Biosciences.

### 2.4. SMRT Sequencing Data Analysis

The data obtained from the PacBio machine were checked for quality using SMRT^®^ Link (v6.0.0.47841) [56]. The subreads bam files produced by SMRT^®^ Link were then converted to CCS (Circular Consensus Sequences) FASTA files using ccs (v3.3.0) from PacBio tools. The ccs FASTA files were then collapsed and counted using the Biostrings package from R [57]. The unique DNA sequences were then queried for potential ORFs and translated to protein sequences using the getorf tool (getorf -minsize 2100 -find 1 -table 0) from EMBOSS [58]. The protein sequences were further collapsed and counted to obtain the final protein frequencies (Counts/Total). Additional filters were applied, in order to remove possible contaminants, prior to the selection of the top clones. The relevant plots were made using various R packages [59].

### 2.5. Recombinant AAV Production

Selected clones that were identified via the SMRT sequencing analysis were synthesized and assembled into a RepCap plasmid backbone, which contained AAV2 Rep and no ITRs. The syntheses were performed by ThermoFisher. Once cloned into the RepCap plasmid backbone, the clones were verified via sanger sequencing at the UTSW McDermott sequencing core. Self-complementary recombinant AAV vectors (scAAV) were produced at the UNC Vector core packaging the CBh-GFP reporter construct (pTRS-ks-CBh-EGFP), as described [3,60]. The final packaged clones were P126-CBh-GFP, P295-CBh-GFP, P72-CBh-GFP, and P2902-CBh-GFP, along with AAV9-CBh-GFP.

### 2.6. AAV/GFP Vector Injections in Mice

All work in mice was reviewed and approved by the UTSW IACUC. The selected clones (P126-CBh-GFP, P295-CBh-GFP, P72-CBh-GFP, and P2902-CBh-GFP) along with AAV9 or vehicle were independently infused via the IT route with 5 × 10^10^ vg per mouse in approximately 8-week-old C57BL/6J mixed male and female mice (n = 3–4), to assess their efficacy in vivo via immunohistochemistry. Three weeks after the injection, the animals were deeply anesthetized via intraperitoneal injection of a standard saline solution containing 2.5% of 2,2,2-Tribromoethanol (avertin, Sigma-Aldrich). Subsequently, the animals were perfused with 1X PBS containing 1 unit/mL heparin for 5 min, and the tissues of interest (brain, spinal cord, heart, liver, spleen, lung, and kidney) were harvested for immunohistochemistry (IHC) against GFP.

### 2.7. Immunohistochemistry and Image Processing

Tissues recovered from mice were immediately fixed in 10% neutral-buffered formalin (NBF) (Sigma-Aldrich). Then, 24 h later the formalin solution was replaced with 70% ethanol (Pharmco, Brookfield, CT, USA). Each tissue was embedded in paraffin and sliced into 5 µm sections using a microtome, and after transferring onto microscope slides, they were treated with 3% hydrogen peroxide (H_2_O_2_) for 30 min at room temperature. Subsequently, two rounds of washing were performed using 1× PBS for the duration of 5 min each. After completing the washing, the slides were treated with 5% goat normal serum (GNS) for 1 h. Following this one-hour incubation, the blocking solution was replaced with another GNS solution, which contained a 1:1000 diluted anti-GFP chicken primary antibody (Aves, Davis, CA, USA, GFP-1020), and the slides were incubated overnight at 4 °C. Another round of washing was performed using tap water and 1× PBS, followed by an incubation period of 1 h at room temperature, with a biotinylated anti-chicken secondary antibody (Vector Labs, San Francisco, CA, USA, BA-9010) in a 1:400 dilution. Furthermore, the sample containing slides underwent additional washing and were incubated with the ABC (Vector Labs PK-6100) at room temperature for 30 min, followed by washing with tap water and 1× PBS. Then, 3,3′-Diaminobenzidine (DAB) (Sigma, D4293) was used to develop the reaction product. Prior to imaging the slides using an Aperio ImageScope, the tissues were counterstained with modified Mayer’s hematoxylin.

### 2.8. Image Analysis and Statistics

The images were analyzed using custom analysis settings in the HALO^TM^ image analysis platform (Halo 2.2, indica Labs). Each region of interest (ROI) was hand drawn on each sagittal organ section, for the analysis of the brain, the heart, the liver, the spleen, the lung, and the kidney. The threshold for each stain was set using positive and negative control images, and the same analysis setting were applied for every image of the same stain. The percent area staining was recorded for each tissue per ROI. The ROIs include the cortex, subcortex (hippocampus + hypothalamus + striatum + thalamus), brainstem, and cerebellum. Analysis was performed with the observer blinded to the treatment group of each sample, and GraphPad 10 software was used to determine the descriptive statistical values. Ordinary one-way analysis of variance (One-way ANOVA) was performed using Graphpad Prism, to compare the tested clones per tissue count with respect to AAV9 with Tukey’s multiple comparisons.

## 3. Results

### 3.1. SMRT Sequencing Enabled High-Throughput Identification of Capsid Shuffled Libraries and Revealed an Input Library Bias Favoring AAV2

To engineer novel AAVs that have enhanced tropism to the CNS after intrathecal (IT) lumbar puncture administration in mice and non-human primates (NHP), we generated a novel shuffled capsid library, termed Original library 1, that incorporated naturally occurring AAVs (AAV1, 2, 6, 8, 9, and Rh10) and additional engineered capsids from our lab and others (AAV2.5, AAV2i8, AAV9.47, Olig001, AAV clone 1, 32, 83, and 114) as parent sequences. Three weeks post injection, tissues were collected and the full VP1 capsid sequence was amplified out of bulk tissue DNA (Figure 1A). To enable high-throughput identification of capsid shuffled AAV variants, VP1 amplicons were SMRT sequenced from specified tissues, sequencing runs were subjected to quality control assessments, and unique protein sequences of VP1 were indexed using a series of bioinformatic filters that converted raw sequencing data into circular consensus sequences (CCS), and then these CCS reads were converted into unique protein sequences (Figure 1A–C). Finally, the recovered VP1 sequences were compared to the parent capsids used for creation of the original library 1, to determine the percentage of parent homology (Figure 1D). Strikingly, SMRT sequencing of the original library recorded over 15 million individual sequencing reads, which identified 300,000 unique DNA sequences and resulted in over 21,000 distinct capsid variants being identified. Interestingly, the sequencing revealed a starting bias in the library towards variants with sequences that resembled mostly AAV1 and AAV2 (Figure 1D). SMRT sequencing was not only able to identify sequences in a high throughput manner, but also demonstrated utility here in the quality control of AAV library construction.

### 3.2. Round 1 of Selection Revealed Capsid Variants with Predominantly AAV1 Sequence Homology in Mice and Clone 83 in NHPs

To identify capsid variants enriched in the CNS, AAV genomes were recovered from the bulk tissue of brain and spinal cord from mice, as well as CNS and peripheral tissues from NHP that were treated with the original library 1. SMRT sequencing revealed more that more than 7 million CCS reads obtained from every sample during the first round of selection (Figure 2A). A robust and consistent read depth was achieved across all samples. This resulted in over 100,000 unique capsid DNA sequences being identified across all tissues (Figure 2B). When converted into unique capsid protein sequences, a wide range between 200 and 43,000 unique capsid amino acid sequences were recovered, depending on the tissue (Figure 2C). Of note, the deep cortex of NHPs appeared to have the least amount of capsid diversification from bulk tissue recovery, with only ~200 unique sequences recovered. Of these ~200 sequences, the composition of the recovered capsids appeared to have highest homology with clone 83; approximately ~99% (Figure 2F). The tissue that appeared to have the highest degree of diversification was the lumbar DRGs in NHP, with 43,000 unique protein sequences (Figure 2C). Further analysis comparing the sequence composition of each set of sequences derived from mice and NHPs revealed that the variants recovered from the brain and spinal cord of mice correlated most with the NHP striatum, hippocampus, deep cortex, brainstem, lumbar spinal cord, cervical DRG, and surprisingly the spleen. The tissues that least correlated with the mouse CNS tissues were the NHP liver and lumbar DRG. As expected, the liver showed large disparities from mice CNS tissues and many NHP CNS tissues (Figure 2D). We further narrowed down our search and selected the top candidate from each sample (Figure 2E). The original library and the NHP liver had a common top candidate known as “P1”. Multiple sequence alignments of P1 and the parent capsids further revealed that P1 had complete homology with AAV2. Therefore, P1 was excluded from further analyses. Even with a bias towards AAV2 in the starting library, a diversity of novel capsid variants were recovered and identified post-selection. Perhaps unsurprisingly, we found that a common variant was most abundant in mice (P126) but it was not abundant in NHPs. Likewise, P19, P2, P295, and P72 appeared to have highest enrichment in the NHP CNS and showed low abundance in mice (Figure 2E). Multiple sequence alignments (Figure 2G) between lead candidates and parent capsids revealed hierarchical clustering based on percent homology (Figure 2F). Finally, pairwise sequence alignments showed that P126 was most similar to AAV1 (~94.6%). P295 showed closest homology with clone 83 (~99%), P72 was most similar to AAV8 (~97.8%), and P2902 was most similar to AAV2 (~99.9%) (Appendix A).

### 3.3. Clones from Round 1 of Selection Exhibited Altered Tropism Compared to AAV9

After the first round of selection in mice and NHPs, we selected four clones that were identified via SMRT sequencing: P126, P295, P72, and P2902. These were packaged with a self-complementary CBh-GFP genome for efficacy validation studies. Lead clone candidates were selected based on their high frequency in mouse brain, mouse spinal cord, and NHP outer cortex, as well as low abundance in the NHP liver. These clones were injected individually in 8-week-old C57BL/6J mice at 5 × 10^10^ vg/mouse via IT injections in parallel with AAV9 and vehicle (Figure 3A). Three weeks post-injection, we examined the CNS and peripheral organs for GFP expression via immunohistochemistry (Figure 3 and Figure 4). We did not observe significant differences in total GFP expression across the entire mouse brain and spinal cord between mice injected with AAV9 and mice that were treated with our lead clone candidates (Figure 3C,E–H). However, interestingly, the assessment of mouse brain subcortical regions revealed significant differences in transduction patterns between AAV9 and three of the four clones (Figure 3D). Altogether, these findings demonstrate the ability of these novel clones to target the central nervous system following IT administration.

Additionally, tissues from the peripheral organs revealed that clones P126, P295, and P2909 had a lower GFP expression in the heart and liver in comparison to AAV9 (Figure 4A–C). Surprisingly, P72 exhibited significantly higher levels of GFP expression in the liver and spleen compared to AAV9 (Figure 4C,D). In summary, these findings highlight that these novel clones had high specificity for the CNS over the liver, except for clone P72.

### 3.4. Round 2 of Selection Showed Differences Compared to the First Round of Selection

After one round of selection, two novel AAV libraries were generated from the recovered capsids. One library, “mouse-derived library”, was generated from capsids recovered from mouse brain samples. The second library, “NHP-derived library”, was generated from capsids recovered from NHP brain samples. These two new libraries were then packaged into their respective capsids and injected IT into the opposing species from that which they were derived. For clarification, the mouse-derived library was injected into NHPs, and the NHP-derived library was injected into mice (Figure 1A). The same paradigm as in the first round of injections was followed for tissue recovery, with the following exception: to enrich for capsid variants that effectively trafficked to the nucleus of transduced cells, viral genomes were amplified from isolated nuclei within the tissues of interest, rather than from the bulk tissue. Comparable read depth was achieved from isolated nuclei. However, we found greater disparities when comparing clones that were enriched in our first round of selection versus our second round of selection (Figure 5 and Figure 6). We found that in comparison to the first round of selections, the input libraries (NHP and mouse-derived libraries) had a library composition that was high in sequence homology to AAV1 followed by clone 83 (Figure 5C,D), whereas in the original library, there was a much higher composition of variants with homology to AAV2, followed by AAV1 (Figure 5A). This highlights the enrichment from directed evolution following our first round of selection, away from AAV2 and toward AAV1 and engineered capsids such as clone 83.

Mice injected with the original library (Figure 5A) appeared to show enrichment in capsids most similar to AAV1 in the brain (Figure 5B) and spinal cord (Figure 5B). After injection with our NHP-derived library, the mouse brain recovered capsids appeared to have higher homology with AAV9 (Figure 5E), one of the least represented capsid variants in the original library 1 and in the NHP-derived library. Furthermore, the variant composition showed more than 50% similarity to AAV1 in the first round of selection, and in the second round of selection, we observed a shift of close to 0% similarity to AAV1.

Lastly, when comparing the composition of NHP recovered variants in the striatum and lumbar spinal cord, we found that the first round of selection variants had most sequence homology to AAV1 (Figure 6C,D). The deep cortex recovered clones (Figure 6B) had most resemblance with AAV2 and clone 83, the lumbar DRG recovered clones (Figure 6E) showed the highest similarity with clone 32 and clone 83, and the outer cortex recovered clones (Figure 6F) showed highest similarities with AAV6 and AAV1. Tissues analyzed after the second round of selection (Figure 6I,J) using our mouse-derived library (Figure 6H) showed a higher diversity of recovered capsids. The deep cortex (Figure 6I) and NHP striatum (Figure 6J) recovered clones showed the highest similarity with AAV1, followed by clone 83 and AAV9.

## 4. Discussion

One of the many problems faced in the field of AAV capsid engineering is the compatibility of viral capsids across multiple preclinical models. To address this major problem, we set out to engineer novel AAV capsids that were compatible in both mice and non-human primates (NHP), for the delivery of genes to the CNS using whole-capsid shuffling coupled with directed evolution.

The analysis and identification of novel capsids initially faced an additional challenge of throughput, created by the limitations of Sanger sequencing. One solution to this was the advancements in NGS, which allowed high-throughput sequencing of recovered capsid sequences but was limited to short reads of ~200 nt. Thus, with the limitations around NGS, the advantages of higher throughput where limited to capsid libraries composed of localized modifications, point mutations, or peptide insertions. Advancements in third-generation sequencing, by means of long hifi reads, have now circumvented these limitations. The read depth and throughput, coupled with long reads, have broadened the opportunity to explore novel methods for library construction and potentially to better understand the biology of AAV. We established a high-throughput method that is amenable to modifications along the entirety of the capsid sequence. Furthermore, we showed that long hifi reads using SMRT sequencing enabled high-throughput analysis for identification of novel AAV clones recovered from animal tissues.

We selected a number of variants from one round of bulk tissue selection to gain an initial assessment of the productivity of the library and selection process. Interestingly, we selected multiple capsids whose sequences were distinct from AAV9 but still achieved a comparable overall efficiency in targeting the brain. There was notable divergence of the recovered variants’ biodistribution patterns relative to AAV9, however. In Figure 2G, we show the multiple sequence alignments of capsids that were used to construct the capsid shuffled library and the selected clones we tested in vivo for validation experiments; and from this analysis, we found long stretches of conservation among all capsids that align to important structural components responsible for the metastability of the virus. Most importantly, the stretches of conservation mostly cover the super conserved parvo coat that VP1, VP2, and VP3 maintain to form the core of the individual subunits. We also saw areas of nuclear localization signals and phospholipase A2 (PLA2) domain that are important for viral trafficking, although biochemical and loss of function studies would confirm these speculative conclusions. Likewise, in regions where we expect higher variability (VRI–VRIX), we found that these regions had more diversity among our selected variants and parent capsids. In vivo, we showed that some of our lead variants (P126 and P295) de-target peripheral tissues, while CNS tissue remains comparable to AAV9. P2902 appeared to underperform AAV9 overall, except in certain regions such as the cerebellum. P72 behaved most similar to AAV9 in both the CNS and peripheral organs, which is interesting given its high similarity to AAV8. The unexpected transduction of the spleen by P72 may be worth further exploration for applications where gene transfer to the spleen is desired. Altogether, these novel clones displayed different properties that might make them suitable for different types of gene therapy applications or as an alternative to AAV9 for CNS targeting.

A high-level overview of the overall composition of recovered clones was initially performed by comparing the recovered capsid variants to that of the parent capsids that were used to generate the original input library. This analysis revealed significant and striking differences in the composition of capsid variants recovered from bulk tissue (1st round of selection) versus isolated nuclei (2nd round of selection). Intuitively, selection of capsids from nuclei would minimize the selection for capsids that bind to the cell surface and inefficiently internalize into the cells and/or traffic to the nucleus. Interestingly, several clones became enriched in the second round of selection, while some of the most dominant clones in the first round of selection appeared to have diminished. This observation suggests that recovering the library from bulk tissue without any specificity for nuclear localization greatly increases the number of “false hits.” Of note, this was observed in a previous directed evolution approach that our group published, where increased rounds of selection lacking specificity for nuclear localization resulted in clones with poor transduction profiles [49]. We thus speculate that many of the variants recovered without the nuclei purification step might be capable of reaching the target tissue efficiently but lack the capability to complete one or more aspect of the full transduction pathway (i.e., bind to cells, internalize, endosome escape, traffic to the nucleus, and/or uncoat). It is important to note that these speculations require further investigation. Furthermore, the second round of selection was from tissue-derived libraries that were recovered from the opposing species. This raises another potential caveat, that the differences in sequencing could also arise from clones that were merely compatible in both NHPs and mice. The full characterization of recovered capsid variants from this library selection remains ongoing, particularly for capsids recovered from the 2nd round (nuclei isolation). For now, this work highlights the utility of long read sequencing in enabling directed evolution utilizing modifications along the entire length of the AAV capsid. It further reinforces the great impact that details of the selection process can impart, such as the recovery of capsid sequences from nuclei rather than bulk tissue. As gene therapies continue to show promise for the treatment of various underlying genetic deficiencies, it is critical that we push the boundaries of capsid technology. Additionally, while we continue to generate novel vectors, it is of utmost importance to maintain safety at the forefront. While we did not observe toxicity of our novel capsid vectors, any application of novel vectors for therapeutic purposes should consider and specifically investigate the possibility of new safety profiles.

## 5. Conclusions

In conclusion, our work highlights the potential advantages of whole-capsid engineering, as well as indispensable methodological improvements for the analysis of recovered capsids, including the nuclei-enrichment step and SMRT sequencing.

## Figures and Tables

**Figure 1 genes-14-01660-f001:**
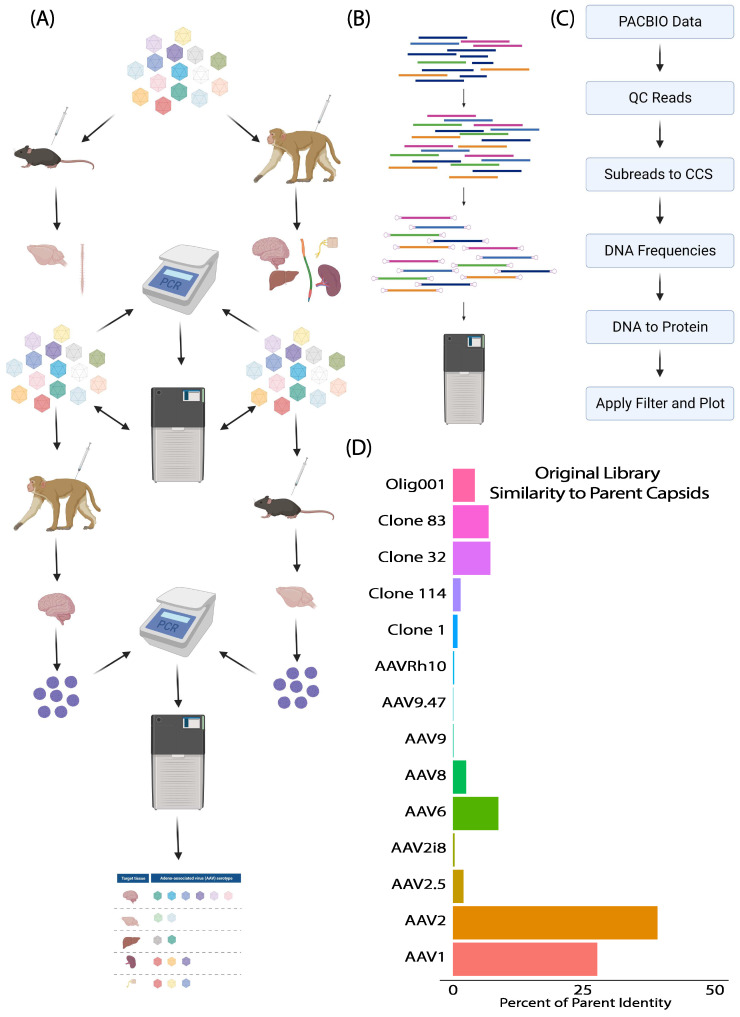
Overview of directed evolution selection and analysis workflow. (**A**) Schematic of AAV capsid shuffled library selection in mice and non-human primates following intrathecal lumbar puncture administration. (**B**) CAP gene amplicons were generated by PCR from input libraries and tissue samples to create SMRT sequencing libraries. (**C**) Sequencing data were subjected to quality controls and de-multiplexed using bioinformatic tools for further novel capsid identification. (**D**) Bar graph of parental library’s variant composition, based on the similarity of variants to parent capsids. The capsids used to construct the library are on the *y*-axis and the percentage of the library most closely matching each parent is on the *y*-axis.

**Figure 2 genes-14-01660-f002:**
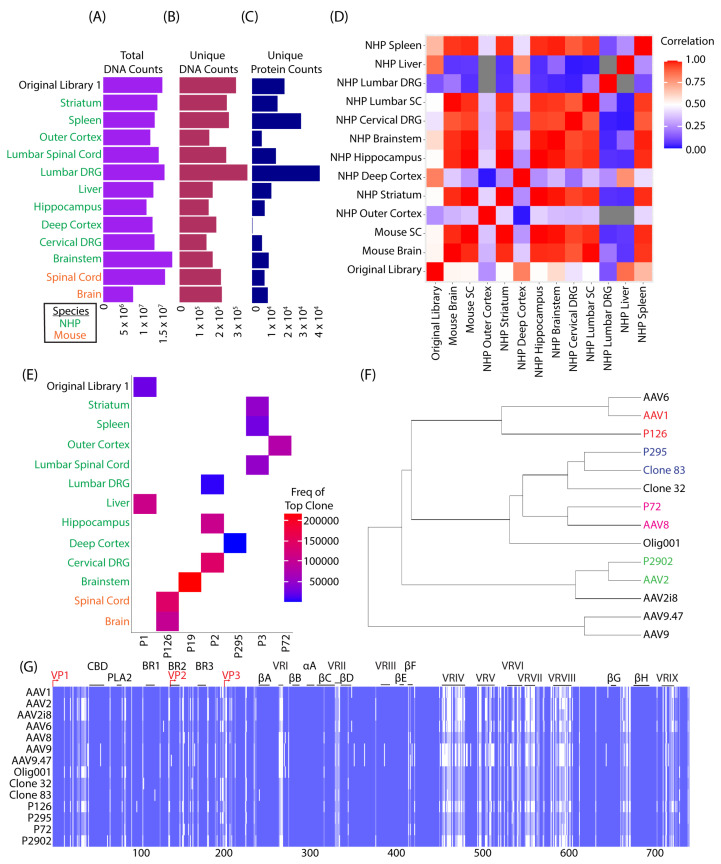
Capsid variant recovery after one round of selection in mice and NHPs. (**A**–**E**) Selections of samples from mice and NHPs are listed along the *y*-axis. (**A**) Summarizes the read depth achieved through SMRT sequencing in “total DNA counts” for the entire CAP gene. (**B**) Filters applied to total DNA reads, to count unique number of DNA sequences followed by (**C**) translation and further filtering to give unique protein sequences (variants). (**D**) The correlation heatmap shows the sample sequence composition of variants (*x* and *y*-axis) compared to one another. The color scale represents the correlation between two samples. (**E**) Frequency heatmap shows top variants (*x*-axis) pulled from each sample (*y*-axis). The color scale represents the total frequency within each sample. (**F**) Dendrogram shows hierarchical clustering of select variants to parent capsids. (**G**) The heat map shows conserved sequence (blue) and non-conserved (white) changes among parent capsids and selected variants. VP1 = viral protein 1; CBD = calcium binding domain; PLA2 = phospholipase A2 domain; BR1–3 = binding region; βA–βH = β sheets A–H; αA = α helix A; VRI–IX = variable region 1–9.

**Figure 3 genes-14-01660-f003:**
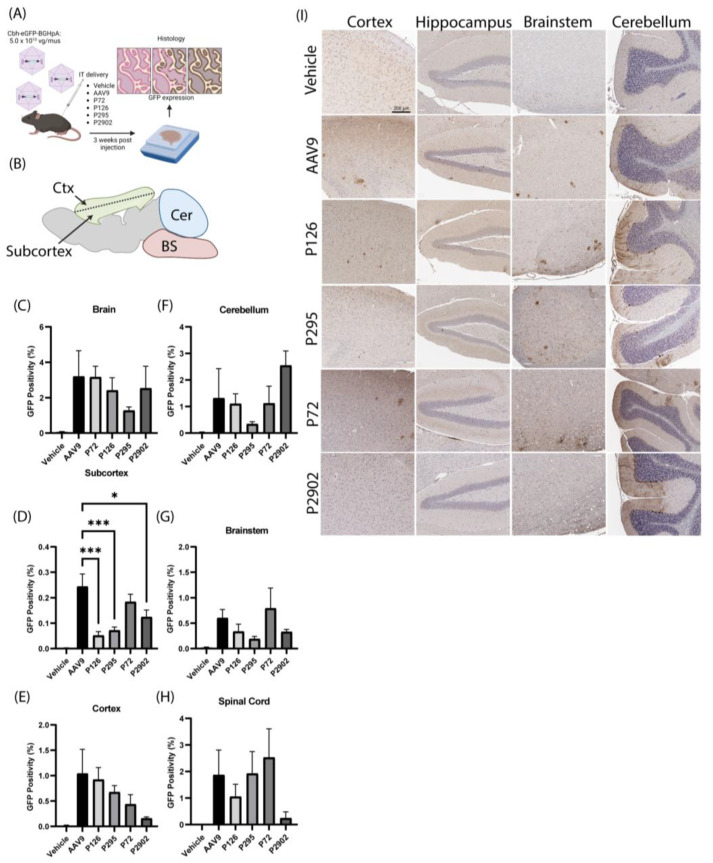
Clones recovered from the first round of selection had altered tropism compared to AAV9 in the CNS of mice. (**A**) Mice were subjected to a single bolus intrathecal (IT) lumbar puncture injection (5 × 10^10^ vg per mouse) of GFP packaged into the respective capsid, or vehicle. (**B**) Brain schematic outlines regions of analysis for GFP positive quantification (Ctx = Cortex; Cer = cerebellum; BS = brainstem). Bar graphs display percentage of GFP staining (*y*-axis) in (**C**) whole brain, (**D**) subcortex, (**E**) cortex, (**F**) cerebellum, (**G**) brainstem, and (**H**) spinal cord (*x*-axis). One-way ANOVA with Tukey’s multiple comparisons; error bars = s.e.m.; * < 0.05; *** < 0.001. (**I**) Representative images show GFP (brown) positive staining and hematoxylin (blue) in the brain. Treatment conditions are listed vertically and brain regions are listed horizontally. Scale bar = 200 µm.

**Figure 4 genes-14-01660-f004:**
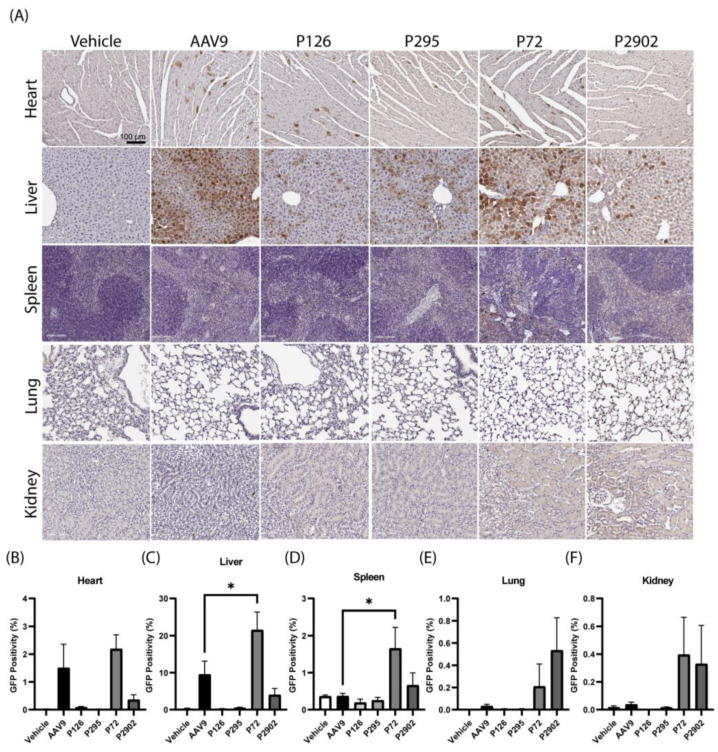
Clones selected from first round of selection had altered tropism compared to AAV9 in peripheral organs. Mice were subjected to a single bolus intrathecal (IT) lumbar puncture injection (5 × 10^10^ vg per mouse) of GFP packaged into the respective capsid or vehicle. (**A**) Representative images of peripheral tissue (horizontal) from each treatment group (vertical). Scale bar = 100 µm. (**B**–**F**) Bar graphs display percentage of GFP staining (*y*-axis) in (**B**) heart, (**C**) liver, (**D**) spleen, (**E**) lung, and (**F**) kidney (*x*-axis). One-way ANOVA with Tukey’s multiple comparisons; error bars = s.e.m.; * < 0.05.

**Figure 5 genes-14-01660-f005:**
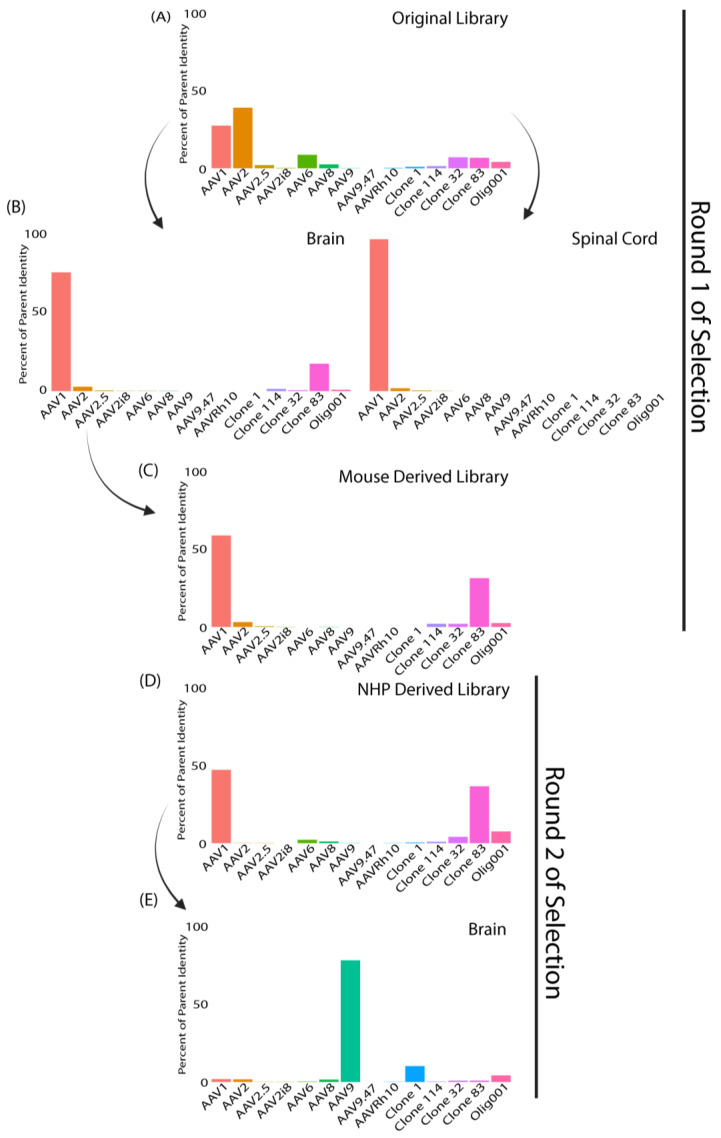
Capsid sequences recovered from nuclei isolation (“Round 2 of selection”) showed major differences compared to capsid sequences recovered from bulk tissue isolation (“Round 1 of isolation”) in mice. (**A**–**E**) Bar graphs display composition of tissue enriched variants. Parent capsids used to construct the library are listed on the *x*-axis and the percentage similarity to the “closest match” parent capsids of variants on the *y*-axis. (**A**) Original input library, (**B**) mouse brain (left) mouse spinal cord (right), (**C**) mouse-derived input library constructed from mouse brain, (**D**) NHP-derived input library after one round of selection constructed from NHP brain regions and spinal cord injected into second round of mice, and (**E**) round 2 of mouse brain.

**Figure 6 genes-14-01660-f006:**
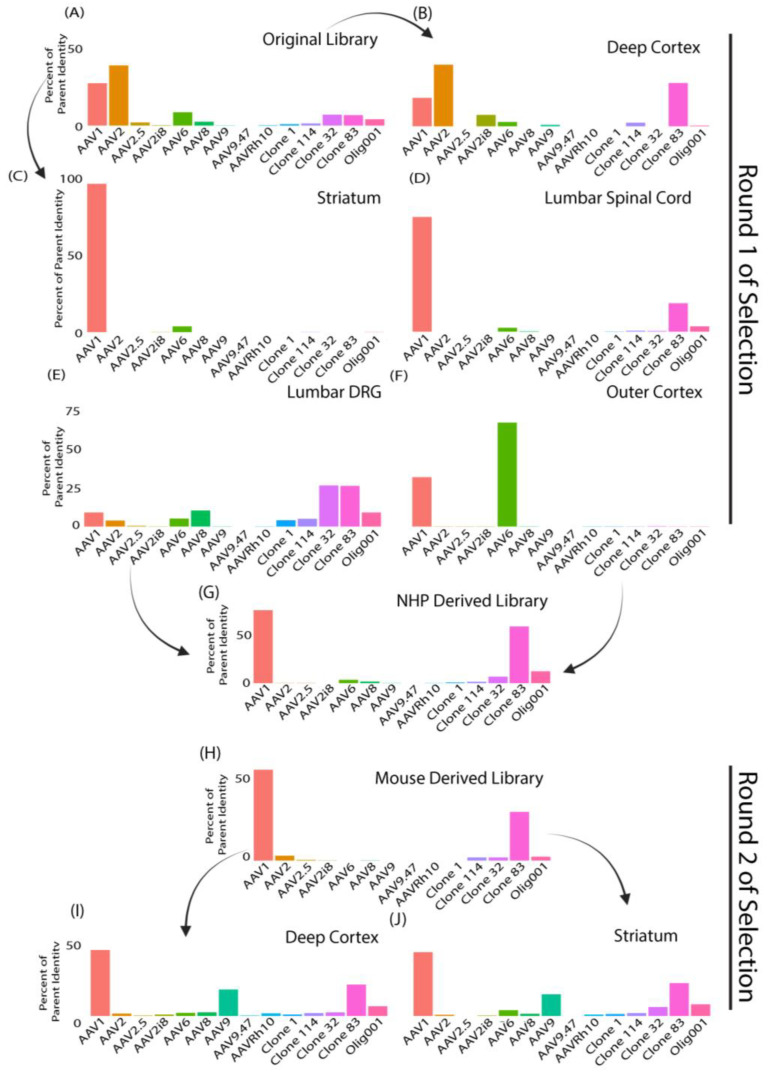
Capsid sequences recovered from nuclei isolation (“Round 2 of selection”) showed major differences compared to capsid sequences recovered from bulk tissue isolation (“Round 1 of isolation”) in NHPs. (**A**–**I**) Bar graphs display the composition of tissue-enriched variants. Parent capsids used to construct the library are listed on the *x*-axis and the percentage similarity to the “closest match” parent capsids of variants on the *y*-axis. (**A**) Original input library. (**B**–**F**) Bar graphs of NHP tissues recovered after one round of selection. (**B**) NHP deep cortex, (**C**) NHP Striatum, (**D**) NHP lumbar spinal cord, (**E**) NHP lumbar DRG, and (**F**) NHP outer cortex. (**G**) Bar graph shows composition of NHP-derived library constructed from pooling NHP brain regions and spinal cord. (**H**) Mouse-derived input library constructed from mouse brain after one round of selection injected into a second round of selection of NHPs. (**I**) NHP deep cortex and (**J**) NHP striatum from the second round of selection.

## Data Availability

Datasets and sequences of capsid variants are available upon request to S.J.G.

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
