# Peer review of "SMRT Sequencing Enables High-Throughput Identification of Novel AAVs from Capsid Shuffling and Directed Evolution"

_genes, 2023, doi:10.3390/genes14081660_

Round 1
Reviewer 1 Report
An interesting study on a couple of fronts - the use of SMRT seq to determine a much larger library of variant cap sequences and, the cross-administration to 2 species to find variants with tropism to neural tissue in both.
The quality of the pictures (histopathology) was poor - low resolution - as were some of the text in the figures.
What was the rationale for choosing the 'top' clones/variants? Why only 4 out of the thousands identified?
Could the authors comment on the data in Fig2G whereby certain regions of Cap appear to be devoid of variants, yet others have many. Presumably critical regions can't tolerate mutation (region 300-400 approx).
The %GFP positives in Fig2 for example are very low - <1% in some cases, yet statistically significant differences are being reported. How many replicates were used? Different animals? How many images of tissue-sections were assessed in each case?
Is that level of transduction even relevant or practically useful for delivery of a therapy?
Otherwise a nice study with some interesting outcomes.
Author Response
- The quality of the pictures (histopathology) was poor - low resolution - as were some of the text in the figures.
We have replaced the histopathology images with lossless images (Figure 3I and Figure 4A-F), which should better preserve the quality of the images. Moreover, we altered the text in the figures to ensure the font resolution is retained.
- What was the rationale for choosing the 'top' clones/variants? Why only 4 out of the thousands identified?
The rationale for picking the lead clones/variants was indicated in the text. In the results section titled Clones from round 1 of selection exhibited altered tropism compared to AAV9 Lines 352-356 “After one first round of selection in mice and NHPs, we selected four clones that were identified via SMRT sequencing: P126, P295, P72, and P2902… Lead clone candidates were selected based on their high frequency in mouse brain, mouse spinal cord, NHP outer cortex, as well as low abundance in the NHP liver.” Analysis of additional interesting capsids beyond this initial set is the focus of ongoing efforts, which is planned for future manuscript submission(s). Those future studies would further investigate the biophysical and biochemical properties that contribute to the altered tropisms.
- Could the authors comment on the data in Fig2G whereby certain regions of Cap appear to be devoid of variants, yet others have many. Presumably critical regions can't tolerate mutation (region 300-400 approx).
We have added lines 476-485 in the discussion to explain this anticipated observation. “In Figure 2G, we show the multiple sequence alignment of capsids that were used to construct the capsid shuffled library and the selected clones we tested in vivo for validation experiments. From this analysis, we find long stretches of conservation among all capsids that align to important structural components responsible for the metastability of the virion. Most importantly, the stretches of conservation mostly cover the super conserved parvo coat that VP1, VP2, and VP3 maintain to form the core of the individual subunits. We also see areas of nuclear localization signals and phospholipase A2 (PLA2) domain that are important for viral trafficking, although biochemical and loss of function studies would confirm these speculative conclusions.”
- The %GFP positives in Fig2 for example are very low - <1% in some cases, yet statistically significant differences are being reported. How many replicates were used? Different animals? How many images of tissue-sections were assessed in each case?
We’ve updated the methods section (lines 227-230) to address this. It is important to note that the doses used in these in-vivo experiments (5 x 1010 vg per mouse) are much lower than what currently is considered therapeutically relevant. We deliberately targeted lower doses where AAV9 was sub-saturating, to make any improvements by capsid variants more readily apparent. Also listed in the same methods section, “C57BL/6J…n3-4…,” indicates the number of mice used per clone. We have also updated lines 257-260, to indicate the number of tissue-section images assessed for these results.
- Is that level of transduction even relevant or practically useful for delivery of a therapy?
Similar to what was mentioned in response for #4. Clones were administered at a much lower dose than what is therapeutically relevant. We aimed to maintain signal below saturation to obtain sensitive comparison among clones and AAV9. The intent was to compare the clones’ performance relative to AAV9, rather than directly demonstrate a therapeutic utility.
Reviewer 2 Report
In this manuscript Casy et al. tried to developed adeno-associated viruses vectors that are compatible across mice and non-human primates for the translation of central nervous system gene therapy applications by generating capsid shuffled libraries of naturally occurring AAVs. The introduction is very comprehensive, providing valuable background information about AAVs and results are in a good fashion.Some improvement should be included.
1. Figure 2 looks a bit confusing. I suggest author reorganize sections and corresponding legend. Following the proposed methodology what are the authors trying to convey; The authors should better explain what were the genes of interest and from what organism they were obtained
2. Can the authors please specify how much sample volume and capsid titer is required for this approach;
3. In Figure 3, the type of statistics analysis, e.g., One-way ANOVA with Tukey’s multiple comparisons, should be explained
4. Can the authors provide details about linear correlation of measured % filled AAV capsids; sensitivity of the method; expected % filled AAV capsids by AUC; LOD and LOQ;
5. A discussion regarding the progress of clinical trial research on adenovirus vectors is clearly not covered in the current version and could be stressed.
6. Conclusions are missing. I wonder if authors would provide a paragraph or some bullet points addressing some major points and challenges of some demanding questions of the discussed area as well as mention future perspectives and possible limitations of this study.
Minor editing of English language required
Author Response
- Figure 2 looks a bit confusing. I suggest author reorganize sections and corresponding legend. Following the proposed methodology what are the authors trying to convey; The authors should better explain what were the genes of interest and from what organism they were obtained
Figure 2 has been updated to better illustrate the major points, including better annotation of panel G to address reviewer #1’s concern. The importance of this is to note that we were recovering and sequencing adeno-associated virus (AAV) capsids via PCR that were injected into both mice and non-human primates as a capsid shuffled library. We show the utility of long hifi sequencing methods to identify mutations that span the full length of the capsid protein, where previous groups have been restricted to short stretches of mutations and peptide insertions within confined residues.
- Can the authors please specify how much sample volume and capsid titer is required for this approach;
We are unclear as to what reviewer 2 is specifically asking about. In terms of the amount of library necessary to inject into animals, we have updated the methods section to reflect this. For the results presented in figures 3-4, the titer and volume information is provided in the methods and each figure legend. In terms of sample for SMRT sequencing, this is based on manufacturer’s specifications for library preparation and sequencing.
- In Figure 3, the type of statistics analysis, e.g., One-way ANOVA with Tukey’s multiple comparisons, should be explained
The methods have been updated to reflect the statistics applied to Figure 3. Lines 266-268.
- Can the authors provide details about linear correlation of measured % filled AAV capsids; sensitivity of the method; expected % filled AAV capsids by AUC; LOD and LOQ;
We appreciate the concern of this comment, however, it is important to note that this approach is to identify novel capsids from a pooled library screen. In follow up in-vivo experiments all clones had similar manufacturability compared to AAV9. Within the scope of our study, all vectors were produced in an identical fashion to assure that their in vivo performance could be compared head-to-head fairly with each other.
- A discussion regarding the progress of clinical trial research on adenovirus vectors is clearly not covered in the current version and could be stressed.
While adeno-associated virus is the primary focus of this study, we are not focused on engineering Adenoviral vectors. The clinical use of AAV vectors is not the focus of this study, but in the introduction we added more background/significance to the reader along with reference to a recent review article from our group. We have added to the end of the discussion a sentence that safety should continue to be of utmost importance while we develop novel AAV vectors. See lines 521-526.
- Conclusions are missing. I wonder if authors would provide a paragraph or some bullet points addressing some major points and challenges of some demanding questions of the discussed area as well as mention future perspectives and possible limitations of this study.
In the Introduction (lines 96-109) and the discussion (lines 508-526), we outline challenges of the current problem and limitations to this study respectively.
Round 2
Reviewer 2 Report
I do not have further revision comments